# Operand Selective Logic Gate Network

## Abstract

We propose Operand-Selective Logic Gate Networks (OSLGN), a symbolic neural architecture that builds differentiable logic circuits via operand and operator selection. Each logic unit dynamically selects two operands from the input and applies one of sixteen predefined binary logic operators, thereby forming a symbolic computation structure that remains trainable through gradient descent. Our operator selection builds upon prior work on differentiable logic gates, while our introduction of operand selection constitutes a novel modular extension. To encourage locally coherent logic formation, we initialize operand selectors with a proximity-based prior inspired by small-world network topology. Specifically, each operand selector is biased toward selecting neighboring input features, allowing the network to efficiently compose local structures and gradually learn long-range dependencies. Experiments on MNIST demonstrate that this initialization improves generalization and stabilizes gradient flow, and we further show that despite modest classification performance, the trained network can be fully converted into compact symbolic logic expressions.

## 1 Introduction

Neural networks have achieved remarkable success across a wide range of tasks, yet their internal representations are often entangled and difficult to modularize. While neural-symbolic reasoning and differentiable logic circuits have made progress toward combining symbolic structure with trainability [12, 11], most architectures still struggle to express logic-based computation in a compositional and scalable way.

In this work, we introduce *Operand Selective Logic Gated Networks (OSLGN)*, a symbolic neural architecture where each layer is a collection of binary logic gates. Each gate selects two operands from the previous layer via a differentiable $\arg\max$ mechanism (using a straight-through estimator), and applies one of 16 predefined binary logic operations (e.g., AND, OR, XOR). While prior works have focused on differentiable operator selection [12], our key contribution is to enable operand-level routing. This allows each gate to learn not only what operation to apply, but also which inputs to apply it to—forming discrete, modular logic circuits.

To encourage the emergence of localized symbolic structure, we initialize operand selectors with a Gaussian proximity bias inspired by small-world connectivity [21, 8]. This inductive prior promotes compositional logic among nearby nodes while still allowing long-range dependencies to emerge through learning.

We further interpret our architecture as a fine-grained form of Mixture-of-Experts [19], where selection happens at the operand level within each logic gate. This facilitates sparse symbolic computation with minimal overhead.

**Contributions.**

Submitted to 39th Conference on Neural Information Processing Systems (NeurIPS 2025). Do not distribute.

- We propose Operand-Selective Logic Gated Networks (OSLGN), a symbolic neural architecture that builds differentiable logic circuits via operand and operator selection.

- We introduce an operand selection mechanism that enables gate-level modularity. While our operator routing builds on prior differentiable logic gates [12], operand selection is independently learned and crucial for symbolic structure.

- We incorporate a proximity-biased operand initialization scheme inspired by small-world networks, promoting symbolic locality and stable training dynamics.

- We show that OSLGN can be fully translated into compact logic expressions post-training, demonstrating the feasibility of distilling a neural network into an explicit symbolic circuit.

## 2 OSLGN Architecture

### 2.1 Overview

Operand Selective Logic Gated Neural Network (OSLGN) is a modular architecture that performs binary logic operations between selected operand features using a learned differentiable selection mechanism. The core design of OSLGN mimics the structure of a logical expression tree, where each logic unit (or layer) selects two operands from the input feature space and applies a logic gate to compute the output.

Each OSLGN layer consists of three components: two operand selectors and one operator selector. The operand selectors learn to identify and extract the most relevant features from the input by applying a sparse one-hot mask generated via an argmax operation over linear projections, smoothed using the straight-through estimator (STE) to maintain differentiability. These selectors ensure that the network can choose which elements of the input vector should interact logically at each step.

The selected operands are then passed to the operator module, which computes a binary logic operation between them. Instead of hardcoding a specific logic gate, the operator is learned as a selection over a set of 16 predefined binary logic functions (e.g., AND, OR, XOR, NAND, etc.). Unlike softmax method from petersen's research[12] an argmax-based weighting is applied to the outputs of all gates, followed by binarization through a custom STE, allowing the model to learn with consisting logic gate identity.

By stacking multiple OSLGN layers, the architecture is able to represent hierarchical logic computations, while maintaining symbolic nature. Unlike standard neural networks, which rely on additive and multiplicative transformations, OSLGN explicitly builds logical reasoning paths via structured operand and operator selection.

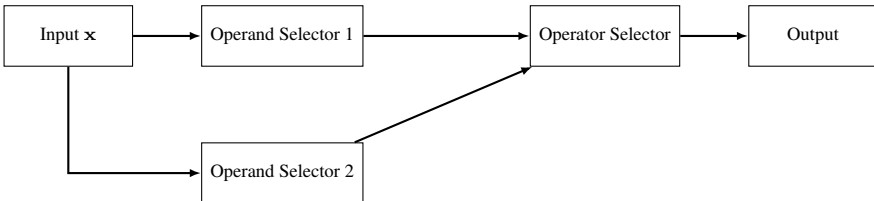

Figure 1: A single OSLGN logic unit. Operand selectors choose relevant inputs, and a differentiable operator module applies one of 16 logic gates.

To improve training stability and promote locally coherent symbolic structures, we initialize the operand selectors using a Gaussian neighborhood prior inspired by small-world network topology [21, 8]. This bias encourages each selector to initially focus on spatially adjacent input features, facilitating modular logic formation while allowing long-range dependencies to emerge via learning.

### 2.2 Operand Selection

In each OSLGN layer, the operand selection modules are responsible for choosing two input sub-components from the feature vector $\mathbf{x} \in \mathbb{R}^d$ that will participate in a logical operation. We refer to

these modules as Operand Selector 1 (OS1) and Operand Selector 2 (OS2). Each selector computes a linear projection over the input and applies a hard selection via the `argmax` function, followed by a one-hot masking operation.

Since `argmax` is non-differentiable, we apply the straight-through estimator (STE) to allow gradient flow during training. Specifically, we subtract the detached projection from the one-hot mask and add back the original projection, enabling gradients to flow through the selected path while preserving the discrete behavior in the forward pass:

$$\tilde{w} = \text{onehot}(\arg\max(w)) - w.detach() + w$$

where $w$ denotes the linear projection weights.

This masked weight vector is then used in a standard linear transformation:

$$\mathbf{a} = \tilde{w} \cdot \mathbf{x}$$

The same mechanism is applied to both operand selectors (OS1 and OS2), producing two selected values $\mathbf{a}$ and $\mathbf{b}$ which are subsequently passed to the operator module.

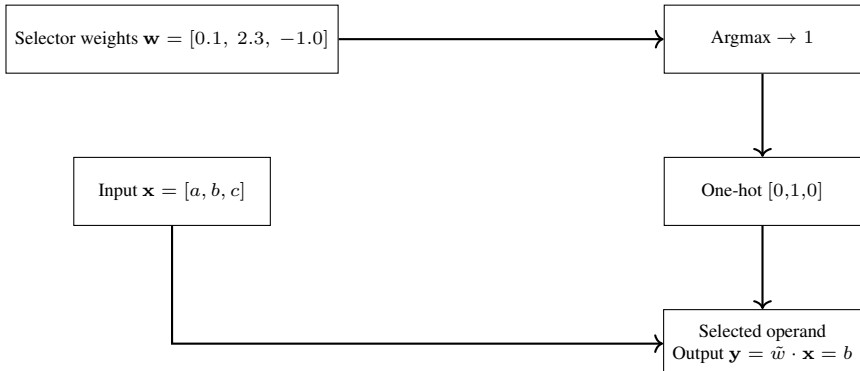

Figure 2: Operand selection with STE. The model uses learnable selector weights $\mathbf{w}$ to generate a one-hot mask that selects a single operand from the binary input $\mathbf{x}$ via a differentiable masking mechanism.

To promote locally structured operand selection, we initialize the selector weights $\mathbf{w}$ using a Gaussian prior centered around each output index. Specifically, for the $i$-th row of OS1 and OS2, the weights are initialized as

$$w_{ij} = \exp\left(-\frac{(j - c_i)^2}{2\sigma^2}\right), \quad c_i = (i + s) \bmod d$$

where $d$ is the input dimension, $s$ is a small center shift (e.g., $s = 0$ for OS1, $s = 1$ for OS2), and $\sigma$ controls the locality. This initialization introduces a topological bias similar to small-world networks [21], favoring local operand pairing early in training.

## 2.3 Operation Selection

While Petersen et al. [12] utilize continuous gate weighting and perform post-hoc logic gate substitution after training, our approach incorporates a straight-through estimator (STE) mechanism that enforces discrete operator selection during training. This results in a train-time quantization effect, where symbolic logic structures are formed during optimization rather than approximated retrospectively.

Specifically, we use a hard one-hot mask over the operator logits with gradient-preserving relaxation:

$$\tilde{\pi} = \text{onehot}(\arg\max(w)) - \text{detach}(w) + w$$

which yields a discrete gate selection in the forward pass, while enabling gradient-based optimization. This tight coupling between learning and logical structure avoids potential mismatch between continuous representations and their final symbolic form.

# 3 Related Work

## 3.1 Logic Neural Networks and Symbolic Computation

Neural-symbolic models aim to integrate logical reasoning into neural networks. Early efforts like DeepProbLog [11] and Logic Tensor Networks [17] incorporated symbolic logic via probabilistic inference or fuzzy semantics, but required predefined logic templates. Neural Logic Machines [4] introduced trainable logical operations, yet imposed rigid operand structures.

Logical Neural Networks (LNNs) [15] extended this line by learning differentiable representations of logical formulas using fuzzy logic operators at the neuron level. While expressive, their logic is embedded in continuous-valued activations. In contrast, our model composes discrete logic circuits via operand and operator selection, allowing symbolic structure to emerge during training. This enables our system to approximate logic expressions in a form directly symbolizable.

## 3.2 Mixture-of-Experts and Modular Routing

Mixture-of-Experts (MoE) architectures enable conditional computation by activating only a subset of experts per input, enhancing scalability and efficiency. Pioneering works like the Sparsely-Gated MoE [19] and Switch Transformer [5] utilize token-level top-k routing. Expert Choice Routing [22] reverses this paradigm, allowing experts to select tokens, improving load balancing.

Recent advancements introduce fine-grained routing and sparse masking techniques. DSelect-k [6] offers differentiable top-k selection without softmax, while BASE Layers [9] employ linear assignment for expert allocation, mitigating expert collapse. Hash Layers [16] provide deterministic routing via hashing functions, eliminating the need for learned gating.

Our approach diverges by implementing operand and operator-level routing within logic gate modules, utilizing argmax with Straight-Through Estimator (STE) for discrete selection. This fine-grained, symbolic routing contrasts with traditional MoE strategies, enabling the construction of symbolic logic circuits within neural networks.

## 3.3 Discrete Selection and Straight-Through Estimators

Training neural networks with discrete operations poses challenges due to non-differentiability. The Straight-Through Estimator (STE) [1] addresses this by treating discrete operations as identity functions during backpropagation, enabling gradient flow through non-differentiable units.

The Gumbel-Softmax trick [7, 10] offers a continuous relaxation of categorical distributions, allowing differentiable sampling. Combining STE with Gumbel-Softmax, the Straight-Through Gumbel-Softmax (ST-GS) estimator performs discrete sampling in the forward pass and uses the relaxed distribution for gradient computation in the backward pass.

Recent advancements, such as Decoupled ST-GS [18], introduce separate temperature parameters for forward and backward passes, enhancing gradient fidelity and training stability.

In our work, we employ argmax-based discrete selection with STE for operand and operator routing within logic gates. This approach ensures the construction of symbolizable logic circuits. While Gumbel-Softmax-based methods demonstrated faster convergence in preliminary experiments, they exhibited instability in training dynamics. Future work may explore integrating advanced techniques like Decoupled ST-GS to balance convergence speed and training stability.

## 3.4 Topological Structures in Deep Learning

Small-world networks, characterized by high clustering and short average path lengths, have been shown to enhance information propagation and convergence in neural networks. Early studies demonstrated that small-world connectivity reduces learning error and accelerates training compared to regular or random networks [21, 14].

Recent architectures, such as SWNet [8], introduce small-world topologies into deep learning models, facilitating gradient flow and feature reuse through long-range connections. These designs improve convergence speed and generalization across various tasks.

Inspired by the efficiency of small-world connectivity, our approach integrates similar principles within logic gate modules. While not implementing a traditional small-world network, we adopt its structural characteristics to enable efficient information flow and modular reasoning. This design choice fosters the emergence of symbolizable logic circuits during training.

# 4 Experiments

Experiments are performed in google colab. Every ipynb is shared via link. MNIST[3] dataset is used.

## 4.1 Ablation Study: Operand Gradient Detachment

To assess the importance of gradient flow through operand selection, we compare two variants of the OSLGN model that share identical architectures but differ in how operands contribute to learning:

- **Model A (STE-enabled)**: Operands are selected using an $\arg\max$ mask with straight-through estimation, enabling gradients to update operand selectors.

- **Model B (Detached)**: The outputs of operand selectors are detached from the computation graph, preventing any gradient flow into operand selection.

Both models were trained on binarized MNIST using the same initialization, optimizer, and training schedule. Table 1 shows that Model A achieves significantly higher performance, reaching a test accuracy of 42.2% compared to just 8.9% for Model B. This suggests that operand selection must remain differentiable for the network to learn meaningful logic compositions.

To ensure logical validity, all outputs of the logic gate layer were enforced to be strictly binary (0 or 1) using runtime assertions during training. The full training script and model code are publicly available via Colab[1], and a complete implementation of the model is included in Appendix A.

Table 1: Operand gradient ablation: detaching operand selection leads to poor performance, confirming its critical role in learning.

| Model | Train Accuracy | Test Accuracy |
|---|---|---|
| Model A (STE-enabled) | 43.3% | 42.2% |
| Model B (Detached) | 10.0% | 8.9% |

## 4.2 Depth Scaling

We investigate how the OSLGN architecture scales with network depth by evaluating variants with 2, 4, and 8 stacked logic layers, each composed of operand and operator selection modules. All models are trained for 50 epochs on MNIST with identical hyperparameters.

Figure 3 shows that the depth-8 model achieves the highest validation accuracy across all depths, slightly outperforming the depth-4 variant. However, the depth-4 model converges more steadily and reaches its peak accuracy earlier, while the depth-8 model continues to improve but with higher variance. The depth-2 model converges quickly but saturates early. These results suggest that deeper OSLGN networks can achieve higher performance, but require more training and exhibit less stability during convergence.

---

[1] https://colab.research.google.com/drive/1ykNB-ezkUh9NhR1eGCZwtzsapPIp_BtM

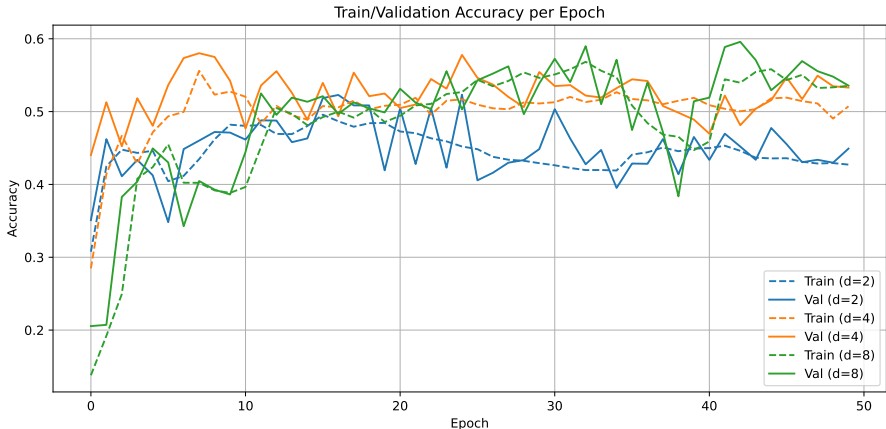

Figure 3: Train and validation accuracy across 50 epochs for OSLGN models of varying depth. Depth-8 achieves the highest validation accuracy, but depth-4 shows more stable convergence.

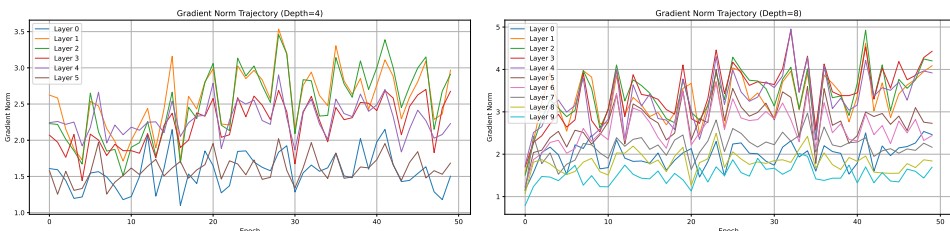

Figure 4: Gradient norm per layer in depth-4(left) and depth-8(right) model. Optimization of depth 8 is less stable, and layer-wise imbalance is more prominent.

Figure 4 show per-layer gradient norms for depth-4 and depth-8 models, respectively. We observe no gradient vanishing across layers; however, the depth-8 model exhibits more pronounced fluctuations and greater disparity between layers, suggesting optimization imbalance in deeper compositions.

Unlike conventional architectures, OSLGN does not employ normalization layers or residual connections, as these mechanisms interfere with the discrete and symbolic structure of binary logic gates. This design choice preserves the model's logic circuit equivalence, but it also introduces training challenges, particularly in deeper networks.

Our implementation[2] is publicly available and supports configurable depth, logging, and visualization, allowing researchers to explore the scalability of logic-based neural circuits further.

### 4.3 Logic Symbol Compression

To evaluate the symbolic expressiveness and compressibility of OSLGN, we extract discrete Boolean expressions from trained models and analyze their logical redundancy. This process is made possible by the discrete nature of the architecture: each layer consists of a set of binary logic gates whose operand and operator selections are recorded during training.

Following training, each class-specific computation path is reconstructed as a Boolean expression using recursive backtracking from the final output node to input variables. These expressions use only `and`, `or`, and `not` operators, forming a fully symbolic circuit.

We apply logic minimization using the `pyeda.boolalg.espresso` package [13], which internally interfaces with the well-known ESPRESSO algorithm [2] for two-level Boolean minimization. For each class expression, we compare the number of logic operators before and after simplification to assess compressibility.

---

[2]`https://colab.research.google.com/drive/1PZhAzfH9bh_2cVn3s_Clmgkj0ezVr3Jw`

**Example: Class 0**

```
[Original]
((((not (x[462] or x[407]) and (x[482] or x[484]))) and
not ((False) and not ((x[578] or not x[363]) or not (x[107]))))) or (False))

[Compressed]
((not x[462] and not x[407] and x[482]) or (not x[462] and
not x[407] and x[484]))
```

While the compressed form often increases in length, this is due to disjunctive normal form (DNF) expansion that enumerates input conditions explicitly [20]. The native OSLGN representation is already compact, demonstrating its structural efficiency.

The remaining expressions, along with full source code and symbolic reconstruction routines, are provided in Appendix C and available at our public Colab notebook:[3]

## 4.4 Effect of Local Operand Initialization

We investigate how incorporating local topological priors into operand selection affects training dynamics and generalization. Inspired by the locality structure of small-world networks [21], we initialize each operand selector (os1, os2) to prefer inputs that are spatially adjacent. Specifically, the $i$-th output of os1 is initialized to select around input index $i$, while os2 is initialized around $i + 1$, using a circular Gaussian kernel. No long-range connectivity is imposed; such dependencies must emerge via learning.

We compare this inductive bias (*local init*) with standard dense initialization (*no init*) using OSLGN models with depth 4 and width 512 on MNIST. All other configurations (loss, optimizer, batch size) are held constant.

**Generalization.** As shown in Figure 5, the model with local operand initialization achieves higher validation accuracy (39.9%) compared to standard initialization (34.9%), despite having lower training accuracy. This suggests that early locality constraints serve as a useful regularizer for symbolic composition.

**Gradient Dynamics.** We further observe that local init results in lower and more stable layer-wise gradient norms (Figure 6). This implies that operand selection in the early training phase is smoother and more modular, avoiding large gradient spikes often caused by arbitrary operand mixing.

**Learned Distant Connection.** Although no explicit small-world graph is instantiated, our design encourages a functional small-world effect: high local clustering (via neighborhood initialization) and potential for long-range connections (via training). This stands in contrast to graph-based small-world CNNs [8], where topology is hard-coded rather than learned. The full code used to reproduce these experiments is available at: [4]

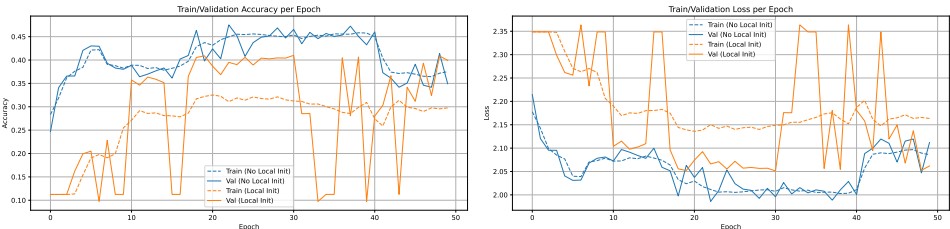

Figure 5: Training and validation accuracy/loss over epochs for OSLGN with and without local operand initialization. Local init generalizes better.

---

[3] https://colab.research.google.com/drive/1VxulftRzLRj1Yg6C-vhmfE6X5jJpxfwJ

[4] https://colab.research.google.com/drive/1iJpgwW6_7oRRcbllWPLOIW9eeZOHcwce?usp=sharing

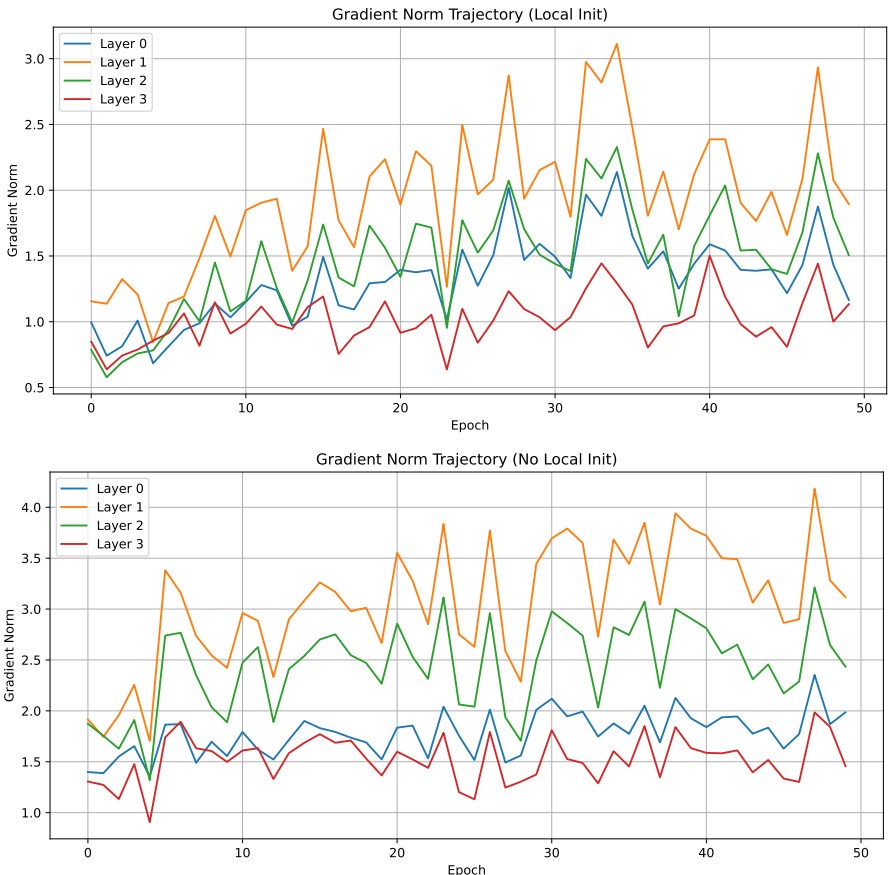

Figure 6: Layer-wise gradient norm trajectories with (top) and without (bottom) local operand initialization. Local init results in smoother gradient flow.

## 5 Discussion

### 5.1 Symbolic Structure and Modularity

Operand-Selective Logic Gate Networks (OSLGN) construct neural models using symbolic logic gate structures. By selecting operands and logic operators at each layer, OSLGN enables a modular architecture that reflects discrete symbolic computations. Unlike prior approaches that approximate logical rules via fuzzy semantics, our model preserves the syntactic structure of logic circuits throughout training.

### 5.2 Discrete Selection Mechanisms

The core mechanism in OSLGN is argmax-based selection with a Straight-Through Estimator (STE), applied to both operand and operator routing. This allows symbolic consistency and gradient-based learning, despite the discrete nature of the computation. While soft selection methods such as Gumbel-Softmax demonstrated faster convergence in preliminary experiments, they require additional stabilization strategies. Future work may explore advanced routing mechanisms to improve convergence speed and training stability.

### 5.3 Topology-Inspired Initialization

Inspired by the information flow properties of small-world networks [21, 8], we initialize operand selectors using Gaussian-weighted connectivity centered on local input indices. While not a true small-world topology, this bias promotes modular selection patterns and improves convergence speed,

as demonstrated in Section 4. This structural prior reflects local coherence and allows long-range dependencies to emerge during learning.

### 5.4 Limitations

OSLGN currently supports only binary logic gates with hard operand selection. This restricts expressivity in tasks requiring multi-bit reasoning or smooth composition. Additionally, the operand selection layers may introduce redundancy: the learned symbolic expressions often use only a subset of the full network's capacity, suggesting overparameterization. Another challenge is training stability during early stages, particularly when the routing distributions are uncertain. While softmax-based selection is viable, convergence strategies for stable symbolic gating remain underexplored.

### 5.5 Future Work

Future directions include incorporating structured routing techniques and annealing strategies, extending the model to multi-valued or temporal logic, and enabling end-to-end symbolic compression during training. Applying the framework to tasks such as program synthesis, structured decision making, or formal verification remains a promising avenue.

## 6 Conclusion

Operand-Selective Logic Gate Networks (OSLGN) represent a step toward unifying deep learning with symbolic reasoning. By learning networks of logic gates through operand and operator selection, OSLGN performs symbolic computation while remaining trainable via gradient-based optimization using the Straight-Through Estimator (STE). We showed that this architecture can be initialized with inductive priors to enhance convergence, trained with discrete routing, and post-hoc translated into compact Boolean expressions.

While classification performance remains limited, the symbolic structure of OSLGN supports modularity and compatibility with digital logic. A key challenge is the relatively large network size required to express symbolic patterns, indicating redundancy or inefficiency in operand routing. Our work lays the groundwork for trainable logic circuits and opens the door for future models that are both compact and symbolic, enabling structured and efficient AI systems.

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

## A  OSLGN Model Implementation

The following PyTorch code defines the core components of the Operand Selective Logic Gated Neural Network (OSLGN) model. Operand selection uses binary masking via $\arg\max$ with a straight-through estimator (STE), and the operator module composes logic gates among 16 predefined binary functions. All logic outputs are strictly binary-valued and verified via runtime assertions.

The logic operator routing and the definition of 16 binary logic gates used in the `Operator` module are adapted from Petersen et al. [12]. Their original implementation is available at `https://github.com/Felix-Petersen/difflogic`. Unlike their soft-selection approach, we apply a straight-through estimator (STE) to enforce discrete operator selection during training, enabling train-time quantization within the logic gate routing mechanism.

## A.1 Operand Selector and Logic Operators

Listing 1: Operand selection and logic gate definitions.

```python
class Operand_selector(nn.Module):
    def __init__(self, x, y):
        super().__init__()
        self.p = nn.Linear(x, y, bias=False)

    def forward(self, x):
        w = self.p.weight
        mask = torch.zeros_like(w).scatter_(1, w.argmax(dim=-1,
            keepdim=True), 1.0)
        masked = mask - w.detach() + w
        return F.linear(x, masked)
```

## A.2 Operator and Logic Layer Composition

Listing 2: Operator routing and composition of logic layers.

```python
def bin_op(a, b, i):
    if i == 0: return torch.zeros_like(a)
    elif i == 1: return a * b
    elif i == 2: return a - a * b
    elif i == 3: return a
    elif i == 4: return b - a * b
    elif i == 5: return b
    elif i == 6: return a + b - 2 * a * b
    elif i == 7: return a + b - a * b
    elif i == 8: return 1 - (a + b - a * b)
    elif i == 9: return 1 - (a + b - 2 * a * b)
    elif i == 10: return 1 - b
    elif i == 11: return 1 - b + a * b
    elif i == 12: return 1 - a
    elif i == 13: return 1 - a + a * b
    elif i == 14: return 1 - a * b
    elif i == 15: return torch.ones_like(a)

class RoundSTE(torch.autograd.Function):
    @staticmethod
    def forward(ctx, input):
        return torch.round(input)
    @staticmethod
    def backward(ctx, grad_output):
        return grad_output

def bin_op_s(a, b, i_s):
    r = torch.zeros_like(a)
    for i in range(16):
        r += i_s[..., i] * bin_op(a, b, i)
    return RoundSTE.apply(r)

class Operator(nn.Module):
    def __init__(self, y):
        super().__init__()
        self.weights = nn.Parameter(torch.randn(y, 16))

    def forward(self, a, b):
        w = self.weights
        mask = torch.zeros_like(w).scatter_(1, w.argmax(dim=-1,
            keepdim=True), 1.0)
        masked = mask - w.detach() + w
        return bin_op_s(a, b, masked)
```

```
405
406  class oslgn(nn.Module):
407      def __init__(self, x, y):
408          super().__init__()
409          self.os1 = Operand_selector(x, y)
410          self.os2 = Operand_selector(x, y)
411          self.op = Operator(y)
412
413      def forward(self, x):
414          a = self.os1(x)
415          b = self.os2(x)
416          return self.op(a, b)
417
```

## B  Depth Scaling Summary Table

To complement the figures in Section 4.2, Table 2 presents a summary of performance metrics for each evaluated depth. We report the final and peak validation accuracy across 50 training epochs, along with final validation loss and training accuracy. This helps assess both convergence stability and generalization behavior. Notably, while depth-4 exhibits both high final and peak accuracy, deeper networks like depth-8 show a wider gap between final and peak accuracy, indicating slower convergence or slight overfitting.

Table 2: Final and maximum validation accuracy, validation loss, and training accuracy for each depth after 50 epochs.

| Depth | Val Acc | Val Loss | Train Acc |
|---|---|---|---|
| 2 | **0.4492** / 0.5232 | 1.7693 | 0.4272 |
| 4 | **0.5330** / 0.5802 | 1.7815 | 0.5073 |
| 8 | **0.5357** / 0.5958 | 1.7304 | 0.5354 |

## C  Symbolic Compression of Learned Logic Circuits

The symbolic logic expressions for each output class were derived from the trained OSLGN model (depth=4, 10 epochs). Each expression was reconstructed by backtracking the operand/operator paths through each logic layer. The code implementation is fully available at:

```
https://colab.research.google.com/drive/
1VxulftRzLRj1Yg6C-vhmfE6X5jJpxfwJ
```

The original expressions from the model and their compressed forms are presented below.

The symbolic logic expressions for each output class were derived from the trained OSLGN model (depth=4, 10 epochs). Each expression was reconstructed by backtracking the operand/operator paths through each logic layer. The expressions are presented below.

### C.1  Class-wise Boolean Expressions

```
[Class 0]

Original
((((not (x[462] or x[407]) and (x[482] or x[484]))) and
not ((False) and not ((x[578] or not x[363]) or not (x[107]))))) or (False))

Compressed
((not x[462] and not x[407] and x[482]) or (not x[462] and
not x[407] and x[484]))
```

```
446
447  [Class 1]
448  Original
449  ((((not (x[355] or x[494])) and not (x[261] or x[438])) and not
450  (((x[177] and not x[290]) or (False)) or ((x[347] and not x[234]) or (False)))))
451
452  Compressed
453  ((not x[355] and not x[494] and not x[261] and not x[438] and
454  x[290] and x[234]) or (not x[355] and not x[494] and not x[261] and not
455  x[438] and not x[177] and x[234]) or (not x[355] and not x[494] and not
456  x[261] and not x[438] and not x[177] and not x[347]) or (not x[355] and not
457  x[494] and not x[261] and not x[438] and x[290] and not x[347]))
458
459  [Class 2]
460  Original
461  ((((not x[372] and x[541]) and not (x[347] and not x[556]))) or
462  ((not (x[319] or x[276])) and ((not x[343] and x[511]))))
463
464  Compressed
465  ((not x[372] and x[541] and x[556]) or (not x[319] and not x[276] and not
466  x[343] and x[511]) or (not x[347] and not x[372] and x[541]))
467
468  [Class 3]
469  Original
470  (not (not (((x[350] or x[322]) and (x[152] or x[179])) or
471  ((x[649]) or (not x[455] and x[564])))))
472
473  Compressed
474  (x[649] or (x[322] and x[152]) or (x[350] and x[152]) or
475  (not x[455] and x[564]) or (x[350] and x[179]) or (x[322] and x[179]))
476
477  [Class 4]
478  Original
479  (not (not ((not (((x[211] and not x[68]) or (False)) or
480  (x[567] or x[127]))) and ((x[401] or x[429]) and (not (x[70] or x[747])))))) or
481  (((not (x[568] or x[595])) and (x[454]))))
482
483  Compressed
484  ((x[68] and not x[567] and not x[127] and x[429] and not x[70] and not
485  x[747]) or (not x[568] and not x[595] and x[454]) or (x[68] and not
486  x[567] and not x[127] and x[401] and not x[70] and not x[747]) or
487  (not x[211] and not x[567] and not x[127] and x[401] and not x[70] and not
488  x[747]) or (not x[211] and not x[567] and not x[127] and x[429] and not
489  x[70] and not x[747]))
490
491  [Class 5]
492  Original
493  (not (((not x[562] and x[517]) and (not x[562] and x[517])) and
494  ((not x[562] and x[517]) and (not x[562] and x[517]))) and
495  (not ((x[246]) and (x[356] and not x[587])) and (not (x[355]) and
496  ((x[191] and not x[276]) or (False)))))
497
498  Compressed
499  ((not x[355] and not x[276] and not x[517] and x[587] and x[191]) or
500  (not x[355] and not x[276] and x[562] and not x[356] and x[191]) or
501  (not x[355] and not x[276] and x[562] and x[587] and x[191]) or
502  (not x[355] and not x[276] and x[562] and not x[246] and x[191]) or
503  (not x[355] and not x[276] and not x[517] and not x[356] and x[191]) or
504  (not x[355] and not x[276] and not x[517] and not x[246] and x[191]))
```

```
505
506  [Class 6]
507  Original
508  (not ((((x[651] or x[658]))) and (((x[651] or x[658])))))
509
510  Compressed
511  (not x[651] and not x[658])
512
513  [Class 7]
514  Original
515  ((((not (x[377] or x[404])))))
516
517  Compressed
518  (not x[377] and not x[404])
519
520  [Class 8]
521  Original
522  (True)
523
524  Compressed
525  (not x[377] and not x[404])
526
527  [Class 9]
528  Original
529  (not (((((x[149] and not x[127]) and not (x[567] or x[127])) or (False)) and
530  not ((x[714] and not x[203]))) or (((((x[149] and not x[127]) and
531  not (x[396] or not x[126])) or (False)) and ((not x[567] and x[711])))))
532
533  Compressed
534  (x[567] or x[127] or not x[149] or (x[714] and not x[203]) or
535  (not x[396] and x[126] and x[711]))
536
```

## C.2 Compression Results

Table 3: Comparison of logic operator counts before and after symbolic compression using PyEDA.

| Class | Original Ops | Compressed Ops | Ratio | Rate (%) |
|-------|--------------|----------------|-------|----------|
| 0 | 13 | 9 | 1.44 | 30.77 |
| 1 | 14 | 43 | 0.33 | -207.14 |
| 2 | 12 | 15 | 0.8 | -25 |
| 3 | 9 | 11 | 0.82 | -22.22 |
| 4 | 17 | 46 | 0.37 | -170.59 |
| 5 | 23 | 48 | 0.48 | -108.7 |
| 6 | 18 | 62 | 0.29 | -244.4 |
| 7 | 2 | 3 | 0.67 | -50 |
| 8 | 0 | 3 | 0 | 0 |
| 9 | 22 | 10 | 2.2 | 54.55 |

These results confirm that OSLGN produces structurally compact logic by design, rather than relying on post-hoc symbolic simplification.

