# OpenReview forum: "Operand Selective Logic Gate Network"
_NeurIPS.cc/2025/Conference — Submitted to NeurIPS 2025_

### Official Review · Reviewer_9bGz · 2025-06-23

**Clarity:** 2
**Significance:** 1
**Originality:** 2
**Rating:** 1
**Confidence:** 4

**Summary:**

The paper introduces Operand Selective Logic Gate Network (OSLGNs), a symbolic neural architecture. OSLGNs, like prior work, replace neurons with a differentiable selection of logic gates. In contrast to prior work, this differentiable selection is a straight-through estimation of an argmax (instead of a softmax), and additionally, the paper similarly adds a differentiable selection of the operands as well, with an initialization scheme. Experiments are performed on MNIST.

**Questions:**

- What is the symbol “j” in the equation below line 88? I don’t believe this is explained.

**Ethical Concerns:**

["NO or VERY MINOR ethics concerns only"]

**Final Justification:**

The key issues remain unresolved:
- The experimental evaluation lacks any competitor.
- The experimental evaluation is only on a very simple dataset (MNIST) and despite this, the results are poor.
- No standard-deviations are reported, no information is given about seeding, and licenses are not explicitly mentioned.

**Limitations:**

Yes

**Quality:**

2

**Strengths And Weaknesses:**

- Strengths
    - The paper has multiple ablation studies.
    - Logic gate networks pose an interesting research direction, and the idea of this approach is interesting as well (selecting not only the operators but also the operands).
    - Use of straight-through estimation is motivated and shown to be useful through ablation studies.
- Weaknesses
    - The experiments are only on MNIST, which is a very easy classification dataset.
    - Accuracy on MNIST is very low (+/- 40%), despite MNIST being a very easy dataset where accuracies near 100% are easily feasible.
    - The experiments do not include any competitors (e.g. previous logic gate networks).
    - The writing can be improved a lot (e.g. line 61 “Unlike softmax method from petersen’s research”, “allowing the model to learn with consisting logic gate identity”, lines 154-155 “MNIST dataset is used.”).
    - Not enough details about the experiments are given in the text. For instance, what is “binarized MNIST” (line 163)? How do you binarize it? While the code is provided, enough details should also be given in the text or appendices, especially if it is necessary to understand the experiments (such as knowing what the dataset is).
    - No mention about which seeds were used (if any) for the experiments.
    - No error bars are reported for the results (e.g. standard-deviation or confidence intervals).
    - I don’t see where the licenses are listed (e.g. for MNIST), despite the authors claiming in the checklist that they list them.

---

> ### Author Rebuttal · Authors · 2025-07-27
>
> Dear Reviewer 9bGz,
>
> We sincerely thank you for your thorough review and for highlighting key areas where the paper can be improved.
> Below, we address each weakness and question in turn.
>
> ---
>
> **Weakness 1: Experiments only on MNIST**
> We agree that the current experiments are limited to MNIST, which is a simple dataset.
> Our primary focus in this submission was to validate the core mechanism of differentiable operand/operator selection in a clean and interpretable setting.
> We are working on stabilizing the architecture further and plan to extend experiments to more complex datasets such as CIFAR-10/100 once the model is sufficiently robust.
>
> ---
>
> **Weakness 2: Low accuracy on MNIST**
> We acknowledge that the classification accuracy is modest (~40%).
> We observed that routing techniques such as softmax and Gumbel-softmax can significantly improve accuracy, but they caused unstable gradients and inconsistent training in deeper models.
> For this submission, we prioritized stability and interpretability over peak accuracy, as we see this work as a foundational step.
>
> ---
>
> **Weakness 3: Lack of competitors**
> We acknowledge that no baseline comparisons with prior logic gate networks were included.
> Because our focus was on validating the operand selection mechanism rather than outperforming existing methods, we did not include competitive results.
> We will clarify this distinction in the revision and plan to add baseline comparisons with prior logic gate network models in future work.
>
> ---
>
> **Weakness 4: Writing and experimental detail**
> We agree that the writing and experiment descriptions can be improved.
> In the revised version we will:
> - Provide a detailed explanation of how "binarized MNIST" is generated.
> - Report seeds and error bars (mean ± standard deviation) for all experiments.
> - Explicitly list dataset licenses (e.g., MNIST, CC BY-SA 3.0) and ensure they are easy to locate in the text.
>
> Regarding error bars:
> We acknowledge their importance and will rerun experiments with multiple seeds and report mean ± standard deviation in the revision.
> In this submission we prioritized broader ablation studies and validating core design choices, so multiple-seed evaluations were deferred to manage computational cost. We agree this information will improve reproducibility and will include it moving forward.
>
> ---
>
> **Question: Symbol “j” in equation (line 88)**
> The symbol “j” simply denotes the operand index in $w_{ij}$ from 2D array of weights to be trained.
> We will clarify this in the revised version, although the formula is standard.
>
> ---
>
> **Summary**
> We appreciate your constructive feedback and suggestions.
> Clarifying the experimental setup, improving reproducibility (multi-seed and error bars), and expanding experiments to more complex datasets and baselines will significantly strengthen the paper, and we are actively working on these improvements.
>
> Thank you again for your careful review.

---

> > ### Comment · Reviewer_9bGz · 2025-08-04
> >
> > I thank the authors for their reply, but my most important concerns remain (e.g. experimental evaluation). Therefore, I will keep my score unchanged and hope the feedback is valuable in improving the manuscript.

---

### Official Review · Reviewer_Lsmi · 2025-07-01

**Clarity:** 2
**Significance:** 2
**Originality:** 2
**Rating:** 2
**Confidence:** 4

**Summary:**

This paper proposes Operand-Selective Logic Gate Networks (OSLGN), a symbolic neural architecture that builds differentiable logic circuits through both operand and operator selection. Unlike prior differentiable logic networks that only support operator choice, OSLGN introduces differentiable operand selection, significantly enhancing modular and compositional capabilities. The architecture is further guided by a Gaussian proximity prior inspired by small-world networks to promote local logic structures and training stability. Although classification performance on binarized MNIST is modest, the network can be fully translated into symbolic Boolean expressions, showcasing strong interpretability and compression potential. The authors provide all code and symbolic tracing routines via open Colab links.

**Questions:**

1. Generalization: Can OSLGN be applied to more complex tasks beyond binarized MNIST, e.g., CLEVR, CIFAR, or logic reasoning datasets?
2. Comparative Baselines: Have you considered comparing to DeepProbLog, LNN, NLM, or other symbolic logic networks?
3. Stability of Operand Routing: Did you observe instability in operand selection during early training? Would annealing help?
4. Symbolic Tracing Efficiency: Can the symbolic expression tracing scale to longer chains or real-world reasoning tasks?
5. Compression Metrics: Beyond operator count, have you considered measuring circuit path diversity or logic depth?

**Ethical Concerns:**

["NO or VERY MINOR ethics concerns only"]

**Limitations:**

The authors include a dedicated “Limitations” section, discussing accuracy gaps, operand redundancy, and future directions. Transparency is commendable.

**Quality:**

1

**Strengths And Weaknesses:**

Strengths:

- Novel architecture introducing operand-level selection in logic gates; a key step forward in symbolic neural computation.
- High interpretability: network outputs can be converted into explicit symbolic Boolean expressions.

Weaknesses:

- Limited experimental scope: only evaluated on binarized MNIST; no comparison with other neuro-symbolic systems.
- Low classification accuracy (~53% max on MNIST), raising questions about general-purpose utility.
- Potential overparameterization: compressed symbolic circuits use only a fraction of available gates.
- Lack of ablation on symbolic generalization: e.g., performance on out-of-distribution logic compositions.
- Writing is dense in places, especially in figures and selector equations, which could benefit from expanded clarity.

---

> ### Author Rebuttal · Authors · 2025-07-27
>
> Dear Reviewer Lsmi,
>
> We sincerely thank you for your detailed and constructive review.
> We appreciate that you highlighted the novelty of operand-level selection and the interpretability aspects of OSLGN. Below, we address each weakness and question in turn.
>
> ---
>
> **Weakness 1: Limited experimental scope**
> We agree that the experimental scope was narrow in this submission.
> Our primary goal was to establish a stable and interpretable operand/operator selection mechanism on a simple benchmark (binarized MNIST) before scaling to more complex tasks.
> We plan to extend OSLGN to datasets such as CIFAR-10/100 once the model becomes more stable, and eventually to more complex structured reasoning datasets when the architecture is sufficiently robust and scalable.
>
> ---
>
> **Weakness 2: Low classification accuracy**
> We acknowledge that classification accuracy remains modest.
> We experimented with routing techniques such as softmax and Gumbel-softmax, which improved accuracy but caused unstable gradient flow and inconsistent training outcomes.
> Our focus in this submission was to build a robust foundation for operand/operator selection. We are currently exploring additional methods to improve stability, including reinforcement-learning-inspired techniques and approaches such as cellular automata (e.g., Google's difflogic CA). While these directions are closely related to our work, they are beyond the scope of the current submission and will be explored in follow-up studies.
>
> ---
>
> **Weakness 3: Potential overparameterization**
> We agree that the compressed symbolic circuits often use only a fraction of the available gates.
> This redundancy arises as the model explores a large operand space early in training, similar to the behavior observed in masked attention mechanisms.
> We plan to refine the operand connectivity mechanism to encourage more efficient gate utilization in future work.
>
> ---
>
> **Weakness 4: Lack of ablation on symbolic generalization**
> We agree this is an important direction.
> As shown in Appendix C, the learned logic is still too coarse to generalize well.
> We plan to add ablations on out-of-distribution logic compositions and explore extensions to architectures such as CNNs to improve generalization.
>
> ---
>
> **Weakness 5: Dense writing and figure clarity**
> We agree that the writing and figures require revision and will improve their clarity in the revised version.
>
> ---
>
> **Question 1: Generalization to complex tasks**
> Yes, we believe OSLGN can be extended beyond MNIST.
> As noted above, we plan to apply it to datasets such as CIFAR-10/100 once the model becomes more stable, and to structured reasoning datasets in the future.
> Currently the model expects binarized inputs, a design choice inspired by byte-level tokenizers.
>
> ---
>
> **Question 2: Comparative baselines**
> We understand the reviewer’s suggestion to include comparisons with symbolic neural network baselines such as DeepProbLog, LNN, and NLM.
> While these systems are valuable, they rely heavily on different modeling assumptions (e.g., pruning and probabilistic reasoning) and are therefore not directly comparable to our approach, which focuses on differentiable operand/operator selection as a modular building block.
>
> ---
>
> **Question 3: Stability of operand routing**
> We appreciate the reviewer’s question and agree that stability in operand selection is important.
> We observed that using adaptive optimizers such as Adam improved stability compared to SGD, effectively acting as a form of annealing.
> We also considered staged training approaches similar to Deep Belief Networks, but these are beyond the scope of the current submission and will be explored in future work.
> We thank the reviewer for this suggestion and will incorporate it into our future investigations.
>
> ---
>
> **Question 4: Symbolic tracing efficiency**
> We agree that the naive complexity of symbolic tracing appears exponential in network depth (O(2^depth)) if all logic paths are enumerated.
> However, by using dynamic programming techniques, we can compute the symbolic expressions with a single forward pass through the network, avoiding the combinatorial explosion.
> We will clarify this in the revision and plan to analyze bottlenecks and further optimize the tracing pipeline in future work.
>
> ---
>
> **Question 5: Compression metrics**
> We currently report operator count and adopted a small-world-inspired initialization strategy to encourage diversity in circuit paths.
> However, we have not directly measured metrics such as path diversity or logic depth. We agree that these would provide a richer view of compression and will explore systematic ways to measure them in future work.
>
> ---
>
> **Summary**
> We appreciate your constructive feedback and suggestions.
> Expanding the experimental scope, clarifying the distinctions from existing systems, and improving clarity will strengthen the paper, and we are actively working on these aspects.
>
> Thank you again for your careful review and valuable suggestions.

---

> > ### Comment · Reviewer_Lsmi · 2025-08-03
> >
> > Thank you for your response, but I maintain my original score, and I hope the feedback can help the authors improve it.

---

### Official Review · Reviewer_c31k · 2025-07-01

**Clarity:** 2
**Significance:** 1
**Originality:** 2
**Rating:** 2
**Confidence:** 3

**Summary:**

This paper proposes Operand-Selective Logic Gate Networks (OSLGN), an architecture that extends differentiable logic gate networks with operand selection. Rather than only learning which operators to apply for each gate, this work also learns the operands that should be fed into those gates. Inspired by small-world connectivity, OSLGN uses a prior that encourages selected operands to be near each other. The experiments demonstrate the importance of gradient flow through operand selection, evaluate the effect of network depth on performance, and justify the use of local operand initialization (which generalizes better). The evaluation also demonstrates that trained networks can be converted into compact logic expressions.

**Questions:**

- What is the motivation for extending differentiable logic gate networks with operand selection? Why did they need to be improved in this way?
- How often are far-away features learned to be selected together? In what cases might this occur?
- What overhead in training time is added by training the operand selector?

**Ethical Concerns:**

["NO or VERY MINOR ethics concerns only"]

**Final Justification:**

While the authors addressed some of my concerns like training time, the evaluation still needs improvement. Since OSLGN does not demonstrate comparable or superior accuracy compared to baselines, I maintain my score.

**Limitations:**

yes

**Quality:**

2

**Strengths And Weaknesses:**

Strengths:

- This paper is related to the promising and active area of logic gate networks research, which can be extremely useful for tasks where fast inference is critical.
- The differentiability of the operand selection was carefully considered, ultimately using the straight-through estimator in place of argmax to ensure differentiability.
- The chosen prior is justified by experiments that show local initialization achieves higher validation accuracy, acting as a regularizer for OSLGN.

Weaknesses:
- It would have been useful if there was more background about the architecture of traditional differentiable logic gate networks. How do these networks select their operands (if not at training time)? Why exactly is this selection sub-optimal?
- The experiments do not appear to use prior work on differentiable logic gate networks [1] as a baseline. It would have been better to empirically evaluate their differences.
- While experiments justify some of the design choices (preserving gradients, using local operand initialization), the validation accuracy is still very low, rarely exceeding 50% on MNIST classification.

[1] Deep differentiable logic gate networks, NeurIPS 2022

---

> ### Author Rebuttal · Authors · 2025-07-26
>
> Dear Reviewer c31k,
>
> We sincerely thank you for your thoughtful and constructive review.
> We are encouraged that you find the topic promising and appreciate the careful questions and feedback you provided, which have helped us better understand how to improve the paper.
>
> Below, we address each weakness and question in turn.
>
> ---
>
> **Weakness 1: Background and operand selection**
> We agree with the reviewer’s comment and will expand the background section in the revision.
>
> Previous research on differentiable logic gate networks focused on **operator selection** and left the operand selection problem to random or brute-force methods. We believe this was because handling both operator and operand selection simultaneously could complicate the model and training.
>
> In our work, we chose to base our approach on this prior model and focused specifically on introducing learnable operand selection. When we used traditional neural networks as operator selectors in preliminary experiments, it was difficult to balance operand and operator selection effectively. This remains an open area we intend to explore further, and we appreciate the reviewer highlighting its importance.
>
> ---
>
> **Weakness 2: Lack of baseline comparison**
> Our work builds directly on the research mentioned by the reviewer, and our focus was not on demonstrating superiority but on **addressing the open problem** left by prior work: learning operand selection rather than relying on brute-force or random choices.
>
> We will make this clearer in the revision and plan to include baseline comparisons with previous logic gate network models in future extensions of the work.
>
> ---
>
> **Weakness 3: Low validation accuracy**
> We prioritized stable training and scalability over peak accuracy for this submission.
>
> We observed that techniques commonly used in Mixture-of-Experts models (e.g., softmax and Gumbel-softmax routing) could significantly improve accuracy, even in smaller models. However, these approaches led to unstable gradients and inconsistent convergence, making it uncertain whether they would scale to deeper architectures.
>
> We are actively working to improve gradient stability so that the model can be extended to larger datasets and architectures.
>
> ---
>
> **Question 1: Motivation and extension**
> We agree that prior work has improved performance by combining differentiable logic gate networks with CNN architectures (e.g., NeurIPS 2024).
>
> Our current focus was to establish a stable basic model following a linear neural network format, which we believe is necessary before extending operand selection to CNNs and other architectures.
>
> While current performance is modest, we expect that future research will allow operand selection to be applied more flexibly in combination with other architectures.
>
> ---
>
> **Question 2: Selection of far-away features**
> As shown in Appendix C1, many results involved selecting far-away features.
>
> We will add tracking and analysis of operand distances in the revision (or a future version) to clarify this behavior, as we agree it is a valuable analysis.
>
> ---
>
> **Question 3: Training overhead**
> Replacing brute-force or random operand selection with learnable selection significantly reduced training time.
>
> While the accuracy is not yet competitive, prior work reported multi-hour training times, whereas our model trains much faster on a single Google Colab T4 GPU (and only used an A100 for the scaling experiment).
>
> We will add a more precise comparison of training time in the revision.
>
> ---
>
> **Summary**
> We are grateful for your constructive feedback and agree that adding background context, baseline comparisons, and expanded experiments will make the paper stronger.
>
> We see this work as a first step toward integrating operand selection into differentiable logic gate networks, and your comments will help us significantly improve the clarity and scope of the paper.

---

> > ### Comment · Reviewer_c31k · 2025-08-02
> >
> > Thank you for your response, but I maintain my original score. The experimental evaluation is still lacking, but I hope the feedback can help the authors improve it.

---

### Official Review · Reviewer_Cpsa · 2025-07-01

**Clarity:** 1
**Significance:** 1
**Originality:** 1
**Rating:** 1
**Confidence:** 5

**Summary:**

The paper investigates the problem of selecting operands from the input for differentiable logic gated networks. To this end, the authors propose a framework named Operand Selective Logic Gated Networks (OSLGN) through a novel modular extension. In particular, this extension consists of two operand selectors, one for each input. These selectors simply computes a linear projection over the input and applies a hard selection via the argmax function, followed by one-hot masking operation. The authors also propose a local initialization scheme using a simple Gaussian prior. The paper provides some empirical evidence on MNIST dataset.

**Questions:**

Please see the Weaknesses. The paper is extremely weak for the moment. There is an additional question:  How is it ensured that the two operand selectors do not always choose the same input, thereby limiting the diversity of operations or comparisons?

**Ethical Concerns:**

["NO or VERY MINOR ethics concerns only"]

**Final Justification:**

I will keep my original score as the rebuttal did not fully resolve my concerns, and I think the paper needs significant improvements.

**Limitations:**

There are a lot of problems for this paper as outlined in the weaknesses. The authors should improve the paper significantly for the next version.

**Quality:**

1

**Strengths And Weaknesses:**

**Strengths**
- The problem of learning to select inputs for differentiable logic gated networks is interesting.

**Weaknesses**

- The paper requires substantial revisions to improve its completeness and writing quality. As a specific example among many, line 154 states: "Experiments are performed in google colab. Every ipynb is shared via link. MNIST[3] dataset is used." Several issues need to be addressed in this sentence. "Google Colab" should be capitalized. The configuration of Google Colab used for the experiments should be specified. A space is needed between "MNIST" and "[3]". What is ipynb? Where is the link?
- The proposed method is totally trivial and significantly below the bar of even a small machine learning conference.
- It is not clear the design of the experiments. For example, how the MNIST data is processed for such kinds of models?
- The experiments are trivial as they only consider simple data like MNIST without any competitors. They are not designed to highlight the main research question which is to learn the operands. In the end, it is not clear what are the advantages of learning operands in the experiments.

---

> ### Author Rebuttal · Authors · 2025-07-26
>
> We thank Reviewer Cpsa for the detailed comments and questions.
> Below, we address each weakness and question in turn.
>
> ---
>
> **Weakness 1: Completeness and writing quality / code availability**
> - All code (4 .ipynb notebooks) was shared via public Colab links at submission. We will ensure that these links are explicitly referenced in the revised version and are easily accessible. You can find them in each experiment section and easily see them from the bottom of the pages.
> - We will also revise minor wording issues (e.g., spacing, capitalization) and add a clear description of the Google Colab environment used. (The code is intended to be used globally, but you might want to know how much time it spent under which environment.)
>
> ---
>
> **Weakness 2: Proposed method is trivial**
> - Our work builds directly on *Deep Differentiable Logic Gate Networks* (Petersen et al., NeurIPS 2022), which has been an influential line of research. Their method focuses on operator selection but relies on brute-force or randomized operand selection and evaluates very large candidate sets (>10k inputs per layer).
> - Our goal is not to replace or outperform prior work, but to explore whether operand selection can be learned instead of chosen manually. We see this as a meaningful step in advancing differentiable logic gate networks.
>
> ---
>
> **Weakness 3: Experimental design clarity**
> - MNIST was binarized into {0,1} pixels as input and trained with standard labels. This simple setting was chosen intentionally to isolate the effect of operand/operator selection without additional confounding factors.
> - We will add more details about preprocessing and the binarization step in the revision.
>
> ---
>
> **Weakness 4: Experiments are trivial (MNIST only)**
> - We agree that broader datasets and baselines will further strengthen the work. Our current priority, however, is to refine the architecture to ensure stable training. We have observed that applying softmax or Gumbel-softmax routing techniques (commonly used in MoE models) can improve accuracy, but they also cause significant instability in gradient flow, leading to inconsistent results.
> - Once we achieve a more robust and stable model, we plan to extend experiments to datasets such as CIFAR-10, CIFAR-100, and other benchmarks beyond MNIST.
> - At this stage, our focus remains on building a reliable operand/operator selection mechanism that can serve as a foundation for future research.
>
> ---
>
> **Question: Operand selector collapse**
> - We understand the concern regarding the possibility of both operand selectors always choosing the same input.
> - Our architecture follows a Mixture-of-Experts (MoE)-style design, which has been extensively validated in the literature and is known to learn diverse routing patterns in practice.
> - In our experiments, we confirmed that gradients propagate correctly through both operand selectors and operator selectors, as shown in Figure 4 of the paper and in the attached Colab notebooks. This indicates that the selectors remain trainable and can adjust their choices during training.
> - While we did not explicitly measure the degree of overlap between selectors, we believe that the training dynamics of the MoE framework make full collapse unlikely. We plan to analyze this aspect in more detail in future work.
>
> ---
>
> **Summary**
> - We see this work as an early step toward addressing an open problem in differentiable logic gate networks: learning operand selection end-to-end.
> - We fully agree that clarifying the paper and expanding experiments to stronger datasets and baselines will strengthen the contribution, and we are actively working toward that.
> - We are grateful for the reviewer’s feedback, which will help us improve the clarity and robustness of the paper.

---

> > ### Comment · Reviewer_Cpsa · 2025-08-04
> > **Official  Comment from Reviewer Cpsa**
> >
> > Thank you for your response. However, I stand by my original score and hope the feedback proves helpful for the authors' revisions.

---

### Note · Authors · 2025-08-15

We thank the reviewers and the area chair for their time, constructive comments, and for facilitating the discussion during the decision process. The feedback provided will be invaluable in improving both the clarity and technical depth of our work.

Our submission introduces **Operand Selective Logic Gated Networks (OSLGN)**, extending prior differentiable logic gate networks by learning both operand and operator selection end-to-end. This work primarily focused on modularity as well as its optimized outcome. We view this as a foundational step toward more expressive and modular neural network architecture.

We acknowledge several important limitations raised in the reviews. The evaluation was restricted to binarized MNIST without broader datasets or baseline comparisons, and the reported accuracy remains modest (~40–53%). These choices reflect a deliberate trade-off to maintain a controlled setting and focus on stabilizing training before scaling the approach.

We also accept that certain reproducibility details, such as the preprocessing procedure, random seeds, error bars, and dataset licenses, were not sufficiently documented. Writing clarity and figure readability can also be improved.

Reflecting feedbacks, we plan to extend experiments to more datasets with competitive accuracy. We will incorporate multiple-seed evaluations with statistical reporting and explicitly document dataset preprocessing and licensing. We are optimistic about applying annealing techniques questioned, because we found promising studies like hierarchical reasoning model.

To address performance limitations, we are exploring reinforcement learning inspired routing to enable deeper architectures without loss of stability. We will also conduct ablations on symbolic generalization, evaluate symbolic tracing efficiency at scale, and adopt richer compression metrics such as path diversity and logic depth.

We are grateful for the detailed feedback, which will guide the next stage of development. We believe that addressing these points will enhance both the practical impact and theoretical significance of OSLGN, and we look forward to sharing stronger results with the community in future work.

---

### Decision · Program_Chairs · 2025-09-17

**Decision:**

Reject

**Comment:**

This paper introduces Operand-Selective Logic Gate Networks (OSLGN), which extend differentiable logic gate networks by incorporating operand selection alongside operator selection. The idea of combining operand and operator routing within symbolic neural architectures is somewhat interesting, and the paper includes efforts toward interpretability by translating networks into symbolic Boolean expressions.

However, after carefully considering the reviews, I believe the submission is below the bar for NeurIPS. The main issues are:

1) While operand selection is presented as the key novelty, the mechanism amounts to a straightforward argmax with a straight-through estimator. This extension feels incremental relative to prior differentiable logic gate networks, and the paper does not convincingly demonstrate that operand selection leads to practical or conceptual advances.

2) Moreover, the empirical evaluation is conducted only on MNIST, a very simple dataset, and even on this task the reported accuracies remain very low (~40–53%). No competitive baselines are included, making it unclear how OSLGN compares to existing differentiable logic networks or neuro-symbolic approaches.

3) Multiple reviewers noted issues with writing, missing experimental details (e.g., how data is processed, what “binarized MNIST” means), and overall polish.

I therefore recommend rejection.